# Role of Defensins in Tumor Biology

**DOI:** 10.3390/ijms24065268

**Published:** 2023-03-09

**Authors:** Lowie Adyns, Paul Proost, Sofie Struyf

**Affiliations:** 1Laboratory of Molecular Immunology, Department of Microbiology, Immunology and Transplantation, Rega Institute for Medical Research, KU Leuven, 3000 Leuven, Belgium; 2Centre for Proteomics, University of Antwerp, 2020 Antwerpen, Belgium; 3Health Unit, VITO, 2400 Mol, Belgium

**Keywords:** defensins, tumor biology, immune cells

## Abstract

Defensins have long been considered as merely antimicrobial peptides. Throughout the years, more immune-related functions have been discovered for both the α-defensin and β-defensin subfamily. This review provides insights into the role of defensins in tumor immunity. Since defensins are present and differentially expressed in certain cancer types, researchers started to unravel their role in the tumor microenvironment. The human neutrophil peptides have been demonstrated to be directly oncolytic by permealizing the cell membrane. Further, defensins can inflict DNA damage and induce apoptosis of tumor cells. In the tumor microenvironment, defensins can act as chemoattractants for subsets of immune cells, such as T cells, immature dendritic cells, monocytes and mast cells. Additionally, by activating the targeted leukocytes, defensins generate pro-inflammatory signals. Moreover, immuno-adjuvant effects have been reported in a variety of models. Therefore, the action of defensins reaches beyond their direct antimicrobial effect, i.e., the lysis of microbes invading the mucosal surfaces. By causing an increase in pro-inflammatory signaling events, cell lysis (generating antigens) and attraction and activation of antigen presenting cells, defensins could have a relevant role in activating the adaptive immune system and generating anti-tumor immunity, and could thus contribute to the success of immune therapy.

## 1. Introduction

Defensins are a family of small cationic peptides containing six cysteine residues connected via three intramolecular disulfide bonds, with a central β-sheet dominating their structure [1,2]. Three subfamilies have been discovered. Lehrer and colleagues identified the first mammalian α-defensins from rabbit granulocytes in 1984. Later, in 1985, they reported the first human defensin sequences, which they initially referred to as antibiotic peptides derived from neutrophils or human neutrophil peptides (HNPs) [3,4]. They also introduced the term ‘defensins’ for the highly related peptides HNP-1 to 3, based on their antibacterial, antiviral and antifungal properties empowering the host defense [4]. Later, HNP-4 was discovered, with the same cysteine backbone as the other myeloid HNPs, but with a slightly different sequence and significantly more hydrophobic amino acids [5]. Only about 2% of the total neutrophil defensin content is HNP-4, which is probably the reason it was overlooked during the discovery of the first three HNPs [6]. Not all mammals have leukocytic α-defensins, as these have only been reported in primates, rabbits and some other rodent species [2]. In this regard, it is important to note that mice lack neutrophil α-defensins [7]. Consequently, researching the role of neutrophil defensins in murine models is only possible in transgenic mice. However, murine intestinal Paneth cells produce α-defensins, which are also referred to as cryptdins [8]. Similar to neutrophil defensins, enteric cryptdins have antimicrobial properties, but the murine cryptdins largely outnumber the group of four human myeloid defensins [9,10]. Around the time of discovery of the mouse cryptdins, two α-defensins were also found to be secreted by human Paneth cells: human defensin 5 and human defensin 6 (HD-5 and HD-6), sharing properties with the myeloid α-defensins [11,12]. HD-5 and HD-6 conclude the group of the six-known human α-defensins. 

The next subfamily that was identified were the β-defensins. Even though their name suggests that those arose after the α-defensins, β-defensins are much older on an evolutionary scale, since α-defensin genes (DEFA) descend from β-defensin genes (DEFB) and both families seem to have evolved from a common pre-mammalian defensin gene [13,14]. Human β-defensins are also more numerous, as almost 40 human β-defensin genes have been identified [15,16]. Interestingly, β-defensins were first detected in the tracheal mucosa of cows [17], and additional β-defensins were purified from bovine neutrophils [18]. It was in 1995, ten years after the first description of α-defensins in humans, that the first human β-defensin was isolated: the human β-defensin-1 or hBD-1 [19]. Since then, more β-defensins have been discovered, and these are all expressed in epithelial and/or mucosal tissues providing antimicrobial protection at sites that are almost continuously in contact with microorganisms [16].

It is also worth noting that θ-defensins exist, which are uniquely expressed by rhesus macaque monocytes and neutrophils. This is the most recently discovered defensin subfamily. They possess a peculiar structure, as θ-defensins are cyclic antimicrobial peptides formed by the ligation of two truncated α-defensins [20].

In general, the structure of human defensins is characterized by a triple-stranded antiparallel β-sheet, held in place by three disulfide bonds [1,21,22] (Figure 1). In α-defensins these disulfide bonds are formed between cysteine residues 1–6, 2–4 and 3–5, while in β-defensins this is between cysteine residues 1–5, 2–4 and 3–6, resulting in a slightly different structure [2].

Soon after their discovery, it was concluded that defensins act primarily antimicrobially, but even at the time it was suggested that defensins may also play a role in inflammation, tissue injury and other processes [26]. Currently, defensins are included in the alarmin family. Alarmins have been thoroughly studied in the context of their role as first-line defenders protecting the host. They are proteins or peptides that act as initiators of a diverse range of immune-related processes [27]. They belong to the broader family of DAMPs (damage-associated molecular patterns) and can be subdivided based on their origin: some are granule-derived, such as defensins, cathelicidin and eosinophil-derived neurotoxin; some have a nuclear origin, such as high mobility group box 1; and some originate from the cytoplasm, e.g., the heat shock proteins [27]. Most granule-derived alarmins are also known as antimicrobial peptides or AMPs, and this subgroup includes defensins.

In normal circumstances, the innate and adaptive immune systems work together to protect us from non-self threats, such as bacteria and viruses. However, although cancer originates from ‘self’ cells, our immune system can recognize and kill malignant cells due to their altered antigenic composition and biologic behavior [28,29]. The genetic instability of cancer cells is the primary source of tumor-specific antigens [29]. In addition, epigenetic abnormalities, changing gene expression, also play an important role in cancer and may cause transcription of genes normally restricted to fetal development during adult life [29,30]. Besides being antigenic, many tumors try to escape from the immune system by creating an immune suppressive environment. It is therefore clear that the interplay between the immune system and tumor cells is complex, and with the rise of immunotherapy the importance of anti-tumor immunity and possible immune escape should not be overlooked. Here we summarize the potential roles of defensins in the tumor microenvironment (TME), as more and more evidence indicates immune-related functions beyond simple antimicrobial activity.

## 2. Direct Effect on Tumor Cells

First, similarly to their antimicrobial effect, α-defensins may have a direct cytotoxic effect on tumor cells. The possible mechanisms of action vary from direct physical interactions with the membrane to the activation of cell death pathways (Figure 2). HNP-1 to 3 induced cell death in A549 cells and Jurkat T-cells associated with mitochondrial injury and other unspecified pathways, with signs of caspase-3/-7 activation [31,32]. Additionally, HNP-1 accumulated in the endoplasmic reticulum before the caspase-3 activation in A549 cells [33]. When recombinant HNP-1 was expressed in A549 cells, it caused significant growth inhibition due to a (probably similar) apoptotic mechanism triggered by the intracellular HNP-1. This anti-tumor activity was also proven in vivo, as tumor cell apoptosis, decreased microvessel density and increased lymphocyte infiltration was seen in mice treated with an eukaryotic expression plasmid encoding HNP-1 [34]. In a biomechanical study with PC-3 cells, Gaspar and colleagues showcased the cytotoxicity of HNP-1, which caused morphological modifications associated with membrane permeabilization [35]. They suggested a two-step cell injury process: first the membrane is permeabilized allowing HNPs to enter the cell, next DNA damage occurs, as HNPs are able to induce single strand DNA breaks [35,36]. Depending on the defensin concentration, membrane disruption can be caused by dimerization of HNP-1, where the hydrophobic side of the dimer faces the lipid chains of the membrane while the polar side forms an aqueous pore, causing cell leakage [37]. The relative ‘selectivity’ of HNPs for cancer cell membranes can be explained by the enrichment in phosphatidylserine in cancer cell membranes, making those more anionic and increasing the chance of interaction with the cationic HNPs [35,38]. High local concentrations of HNP-1 (≥10 µg/mL) also exerted cytotoxic effects on keratinocytes, primary epithelial cells and fibroblasts [39,40]. Similarly, exposure of oral squamous cell carcinoma cells to high concentrations of HNP-1 resulted in an oncolytic effect [41]. α-defensins purified from neutrophils also showed a synergistic anti-tumoral effect when administered with the antibioticum, nisin, by inducing apoptosis on prostate (PC-3) and colorectal cancer (HCT-116) cell lines [42]. Not only are the neutrophil derived defensins cytotoxic, but α-defensin 5, which is mainly expressed in Paneth cells, affects tumor cell viability. Colon cancer cell proliferation and colony formation capacity were significantly decreased by DEFA5 overexpression. In nude mice, overexpression of this gene suppressed tumor growth. The mechanisms behind its tumor suppressive effect involves phosphoinositide 3-kinase, as DEFA5 binds directly to its signaling complex, leading to delayed cell growth and metastasis [43]. Interestingly, hBD-1 is able to alter human epidermal growth factor receptor 2 (HER2) signal transduction and urine-derived hBD-1 was able to suppress bladder cancer growth [44]. In human hepatocellular carcinoma (HCC) cell lines, the expression of hBD-1 is dramatically downregulated, and rescuing its expression effectively suppresses cell proliferation and colony forming ability. When tested in a nude mouse hepatocellular carcinoma model, hBD-1 expression inhibited tumor growth by inducing protein degradation and endoplasmic reticulum (ER) stress, and this subsequently activated the c-Jun N-terminal kinase (JNK) pathway, which mediated the inhibitory effect of hBD-1 [45]. Furthermore, hBD-2 and -3 were reported to contain an oncolytic motif that binds to phosphatidylinositol 4,5-bisphosphate. This interaction is critical for mediating cytolysis of tumor cells, and experiments with hBD-2 showed that the defensin killed tumor cells via acute lytic cell death instead of apoptosis [46,47]. Other research confirmed this, as A549 adenocarcinoma cells treated with hBD-3 displayed immediate cell membrane damage.

Furthermore, Defb14, the mouse homolog of hBD-3, was able to significantly diminish tumor growth of Lewis lung carcinoma in mice when continuously infused [48]. hBD-5 also showed promising in vivo anti-cancer efficacy in a 1,2-dimethylhydrazine induced colon cancer model. A decrease in tumor parameters, aberrant crypt foci and an increase in apoptosis rate were observed, concomitantly with tumor infiltration by neutrophilic granulocytes. Colons of hBD-5-treated mice even revealed a restoration of the normal architecture. hBD-5 binds more to cancerous cells, due to the altered fluidity of their cellular membranes, ultimately not affecting healthy host cells [49]. Even though θ-defensins are only found in certain Old-World monkey species, their tumor cell killing capacities cannot be ignored. Serine-rich θ-defensin analogues showed more cytotoxicity towards breast cancer cell lines than towards normal mammary epithelial cells. More importantly, the analogues had a synergistic effect on cisplatin and doxorubicin hydrochloride treatment of a triple-negative breast cancer cell line [50]. Not only animals have antimicrobial peptides. PvD1 is an example of a defensin-like antimicrobial peptide found in the common bean plant (*Phaseolus vulgaris*), which also seems to have some direct anti-cancer activity [51,52]. The peptide had a different effect on normal compared to tumor cells, as it was able to reach the interior of breast tumor cells and to induce apoptotic events. Similar to other defensins, PvD1 interacts with membranes and sometimes causes disruptions. In addition, PvD1 modulated cell-to-cell adhesion. This could be an interesting way to prevent cancer cell adhesion to healthy tissue and suppress metastatic spreading [52]. 

## 3. Immune-Related Functions Relevant in the Tumor Microenvironment

Gaining insight in the tumor microenvironment is crucial to improve the understanding of cancer-immune system interactions, and the effects of immune therapy thereupon. To start off, α-defensins have chemotactic properties (Figure 3). The α-defensins HNP-1 to 3 have a chemotactic effect on CD4+/CD45RA+ naive and CD8+ T cells, monocytes, immature dendritic cells (imDCs) and mast cells [53,54,55,56,57,58]. Chemoattraction of T cells and imDCs by HNPs can be inhibited with pertussis toxin, suggesting that this is mediated by a Gαi protein-coupled receptor [57]. The β-defensins also chemoattract subsets of immune cells (Figure 3). hBD-1 to 3 were able to attract cells transfected with the C-C chemokine receptor 6 (CCR6), which suggests that Th cells, Treg, imDCs and neutrophils could be recruited by hBD-1 [59,60,61]. Immunoglobulin-fusion proteins of hBD-2 and hBD-3 also interact with C-C chemokine receptor 2 (CCR2) and are able to attract CCR2+ peripheral blood mononuclear cells (PBMCs) [62]. Interestingly, CCR2 and CCR6 are important receptors for the recruitment of antigen-presenting cells to inflamed tissues, such as tumors [63]. Synthetic and skin-derived hBD-2 chemoattracted a memory subset of human peripheral blood T (CD4+/CD45RO+) cells and CD34+ progenitor-derived DCs [59]. Moreover, hBD-2 attracted mast cells in a G protein-phospholipase C (PLC) mediated way [64]. Both hBD-3 and 4 induced migration of monocytes, but not neutrophils or eosinophils [65,66]. For both peptides this happens, interestingly, without mobilization of intracellular Ca^2+^ [66].

Defensins are not only able to attract immune cells, they also exert direct activating effects (Figure 4). HNPs trigger degranulation of mast cells, causing release of histamine, together with a variety of cytokines and chemokines [67]. The large spectrum of mediators released from mast cells are able to recruit DCs, eosinophils and neutrophils while concomitantly increasing the endothelial permeability [68,69]. Besides chemoattracting monocytes, HNPs also increase the monocyte-endothelial interactions, facilitating their extravasation and influx into the tissue [70]. The oxidative burst of monocytes is also increased when HNPs are present. Treatment of airway epithelial cells with HNP-1–3 enhanced CXCL8 and CXCL5 expression and release [71,72], and both these CXC-chemokines are known to chemoattract neutrophils [73,74]. In plasmacytoid DCs (pDCs), HNP-1 has been reported to induce the pro-inflammatory type I interferons (IFNs) and IL-6, together with an enhanced TLR9 activation [75,76]. Type I IFNs have an important task in (cancer) immunosurveillance, increasing DC maturation, DC-CD8+ T cell cross-priming, T cell cytotoxicity, natural killer (NK) cell cytotoxicity, macrophage inflammation and more [77]. Additionally, in LPS-primed macrophages, HNP-1 can promote pyroptosis and increase IL-1β release through inflammasome activation via the P2X7 receptor [78]. HNP-1 was reported to increase oxidative stress and autophagy-related genes, and with this enhanced endocytosis more antigens could be processed and presented [79]. Through the P2Y6 receptor signaling pathway, HNP-1 suppresses neutrophil apoptosis, prolonging their lifespan [80]. Jurkat T cells, when exposed to HD-5 and HNP-1, underwent changes in cell shape and rearrangement of the actin cytoskeleton. Both peptides also increased cell adhesion to fibronectin, and consequently Caco-2 cells treated with HD-5 or HNP-1 retained significantly more T cells than untreated cells [81]. Co-cultures consisting of non-small cell lung cancer (NSCLC) cells and peripheral blood mononuclear cells were shown to produce more IFN-γ when HNP-1 was added [82]. Since IFN-γ is an important activating immune mediator, its increase in the tumor microenvironment can play an important role in cancer immunity. IFN-γ is linked to T cell and NK cell migration, loss of function and suppression of Treg cells, maturation of NK cells and an increase in priming and antigen presentation by antigen presenting cells [83]. An enhanced influx of immune cells, an increase of antigen presentation and more antigen release due to dying tumor cells can be expected to result in a stronger anti-tumoral immunity. Thus, human α-defensins both directly and indirectly increase the immune activity where they are expressed or released, including in the tumor microenvironment.

Another interesting interaction that could be relevant in the tumor microenvironment is the interaction between defensins and complement factors, though conflicting interaction models have been proposed. First, binding of HNP-1 to activated complement factor 1 (C1) and C1 inhibitor (C1i) complexes has been reported. Initial inactivation of C1 by C1i is required, as C1i likely changes the conformation of C1 to allow defensin-binding. Those C1-C1i complexes can bind and clear defensins at their physiological concentrations in human plasma by stimulating cellular uptake of the HNP-C1-C1i complexes, thus serving as a defensin recycling method. This uptake could prevent unwanted cytotoxic injury to the surrounding tissue [84]. However, a later report described that HNP-1 specifically binds to the collagen-like stalks of C1q, inhibiting its hemolytic activity. In this latter report the uptake-recycle hypothesis is contested, and it is suggested that HNP-1 functions as a complement inhibitor in the microenvironment of the neutrophils that release the defensin [85]. Alternatively, when HNP-1 to 3 were fixed to a plate, C1q could bind, triggering the classical complement pathway and causing C4b binding to the plate [86]. Additional research in an ELISA based system claimed that HNP-1 in solution binds both mannose-binding lectin (MBL) and C1q. Those results were interpreted as inhibition of both the classical and lectin pathway [87]. Hence, there appears to be a difference in mode of action between surface-bound or liquid-phase HNP-1. Surface-bound HNP-1 may act as opsonin by binding and activating complement, while liquid phase HNP-1 may be adsorbed by serum proteins, such as C1-C1i complexes or might act as a complement inhibitor [84,85,86,87]. HNP-1 is not the only defensin that binds C1q, as hBD-2 was recently described to inhibit the classical complement pathway by binding strongly to C1q, protecting the inflamed tissue against uncontrolled activation of the complement system [88]. However, hBD-2 did not affect the alternative pathway. It is clear that the interaction between defensins and complement factors needs more investigation. If defensins would actually activate the complement system, this would result in more inflammation in the tissue where they are released, causing more tissue damage and possible antigen release from tumor cells. However, if defensins inhibit the complement system, it may be a way to protect the tissues, but this implies that tumor cells could also benefit from this protection.

β-defensins have a wide range of effects on (immune) cells as well (Figure 5). Next to induction of migration, hBD-2 can also trigger histamine release, intracellular Ca^2+^ mobilization and cyclooxygenase-1 mediated prostaglandin D2 production by mast cells [89]. These effects could be abolished by pertussis toxin, suggesting that hBD-2 mediates these effects in a Gα_i_ dependent manner [89]. In a tumor model based on harvested macrophages and breast cancer cells from Swiss albino mice, hBD-2 treatment of the macrophages increased the expression of inflammatory cytokines (IFN-γ, interleukin 1α (IL-1α) and tumor necrosis factor α (TNF-α)) and chemokines (CXCL1, CXCL5 and CCL5). In co-cultures of hBD-2 stimulated macrophages and tumor cells, a higher proportion of dead tumor cells was detected compared to co-cultures with unstimulated macrophages [90]. When hBD-3 was added to human airway smooth muscle cells, a significant increase of the pro-inflammatory chemokine CXCL8 was measured in the culture supernatant. This was not seen when hBD-1 was added. Furthermore, CXCL8 production was mediated by CCR6 and through the extracellular signal-regulated kinase 1/2 mitogen-activated protein kinase (ERK1/2 MAPK), c-Jun N-terminal kinase mitogen-activated protein kinase (JNK MAPK) and nuclear factor-κB (NF-κB) signaling pathways. Moreover, hBD-3 simultaneously induced apoptosis of the airway smooth muscle cells, mutually regulated by mitochondrial reactive oxygen species [91]. hBD-3 was also reported to act on the Toll-like receptors (TLRs) TLR1 and TLR2 of antigen-presenting cells, activating the NF-κB pathway in monocytes and mature DCs, ultimately enhancing gene expression of inflammatory mediators [92]. In mouse Flt-3-induced dendritic cells and human peripheral blood mononuclear cells, hBD-3 could also clearly interact with TLR9 to enhance the response to bacterial DNA [93]. In a model with human Langerhans cell-like DCs (LC-DCs), it has been shown that hBD-3 induced maturation, increased CCR7 expression and T helper type 1 (Th1) skewing function in these LC-DCs. Furthermore, hBD3-stimulated LC-DCs induced proliferation and IFN-γ release by naive human T cells. Similar effects were observed with primary migratory DCs from human skin explants, which strengthens the hypothesis that hBD-3 contributes to the integration of innate and adaptive immunity as an endogenous adjuvant [94].

All these aspects together suggest that defensins have, in addition to a role in the innate immune system, a role in activating the adaptive immune system by increasing pro-inflammatory signaling and promoting activation of antigen presenting cells. This is definitely relevant in anti-tumoral immune responses and could play a role in immune therapy success.

## 4. Proof of Immune-Stimulating Effects in (Animal) Models

Several studies substantiate that HNPs are potent immunoadjuvants in murine models. Intranasal delivery of these defensins together with ovalbumin (OVA) enhanced systemic immunoglobulin G (IgG) responses through CD4+ Th1- and Th2-type cytokines and promotion of B and T cell interactions [95]. In another study, intraperitoneal co-injection of HNPs in mice upregulated antigen-specific Ig production, confirming that defensins released by neutrophils may contribute to both innate and adaptive immune responses [96]. In established CT26 colon cancer and 4T1 breast cancer mice models, intratumoral expression of a secretable form of HNP-1 caused chemotactic and activating effects on imDCs, paired with significant tumor growth inhibition, increased CTL infiltration and an increase in specific humoral immune responses [97]. Administration of HNP-2 in transplanted human papillomavirus positive (HPV+) keratinocytes that formed neoplastic lesions in mice reverted a frequently seen immune alteration observed in cancer development by recruiting intravenously injected human DCs to the tumor [98]. 

Additionally, DNA immunizations with fusion constructs containing hBD-2 could elicit potent humoral immune responses against an otherwise non-immunogenic lymphoma antigen in mice, resulting in a successful protective and therapeutic anti-tumor immunity [62]. When HNPs or hBDs were co-administered intranasally with OVA in C57BL/6 mice, HNP-1 and hBD-2 induced significantly higher OVA-specific IgG, lower IgM and lower IFN-γ, while HNP-2 induced low IgG and higher IgM, and hBD-1 induced higher IgG, higher IgM and higher IL-10 [99]. Subcutaneous administration of hBD-2 in mice showed good tolerability and hBD-2 rapidly entered the bloodstream, allowing for systemic effects if therapy via this method ever becomes an option [100].

These immune activating or anti-tumoral effects of defensins are not only observed in mice. In Drosophila, for example, when the defensin gene (def) was deleted from both alleles by CRISPR/Cas9, dlg tumor growth was not restrained anymore and less tumor cell death was seen. This fruit fly defensin is normally expressed in immune tissues, and it seemed to discern between healthy and tumor cells by the aberrant phosphatidylserine expression in tumor cells, confirming a theory mentioned before. The TNF factor homolog Eiger was required for this phosphatidylserine exposure on tumor cells, allowing recognition by the defensin [101]. Yang et al. speculated that defensins possibly bind to antigenic microbial membranes after killing the microbes, forming defensin-antigen complexes. Complexes like this could then be internalized by imDCs, possibly in a defensin-receptor mediated way, improving the antigen presentation of these cells. Considering this, such complexes could also be created with other (tumor-related) antigens, with a positive effect on the overall immunogenicity of tumors [102]. Moreover, α-defensins have also been reported to inhibit angiogenesis, for example in pathologic retinal neovascularization and in different angiogenesis tests such as the sprout formation assay and chicken chorioallantois membrane (CAM) assay [103,104].

## 5. Expression of Defensins in Cancer

When looking more globally at the expression of defensins in cancer, there seems to be a differential expression between normal and tumor tissues for both α- and β-defensins. Integrative analyses of copy-number profiles of more than 10 000 patients across 31 cancer types from the Cancer Genome Atlas confirm the link between interferon (IFN) activity, defensin genes and immunotherapy resistance. Type-I IFN and α- and β-defensin genes were homozygously deleted in almost 20 cancer types with high frequencies, and the homozygous deletion of IFNs was linked to a significantly shortened overall and disease-free survival time in certain cancer types. Especially the ratio of homozygous deletion of IFNs (HDI) and defensins (HDD) was of interest, as in the most common cancer types the HDI/HDD ratios were significantly higher compared to the rare cancer types, suggesting that HDI/HDD is a molecular mechanism involved in the carcinogenesis of these cancers. This HDI/HDD ratio was the highest in brain lower grade glioma, glioblastoma, pancreatic adenocarcinoma and mesothelioma. In contrast, cancers with a high HDD/HDI ratio are prostate adenocarcinoma, colon adenocarcinoma, rectum adenocarcinoma, liver hepatocellular carcinoma and urine carcinosarcoma. Between these two groups are the cancers with a more equal amount of HDI and HDD, and these include bladder urothelial carcinoma, lung adenocarcinoma, lung squamous cell carcinoma, breast invasive carcinoma, ovarian serous cystadenocarcinoma, head and neck squamous cell carcinoma, skin cutaneous melanoma, esophageal carcinoma, sarcoma and stomach adenocarcinoma [105]. Gene expression analyses showed signs of activation of oncogenic and cell-cycle pathways and repression of genes involved in immune response pathways when there was a homozygous deletion of IFN and defensin genes, suggesting a certain tumor-suppressing effect of defensins that is eliminated when deletion occurs [105]. This immune stimulating effect varies between defensin families (α and β), but differences are also seen inside the families.

Since HNP-1 to 3 are increased in a high variety of tumor types (lung, oral squamous cell, muco epidermoid, bladder, renal colorectal, breast, head and neck cancer) [106], it is definitely interesting to unravel their immunological functions in these environments. In pretreatment tumor biopsies of non-small cell lung cancer patients, mass spectrometry imaging revealed that HNP-1 to 3 can have a role as additional prospective biomarkers for anti-PD-(L)1 immunotherapy response, as the addition of these biomarkers predicted the therapy response better than PD-L1 expression alone [82]. However, other α-defensins are also relevant in cancer. The expression level of α-defensin 5 (DEFA5) was drastically downregulated in human gastric cancer. When overexpressing DEFA5 again in gastric cancer cell lines, this strongly diminished cell proliferation and the ability to form colonies. This suppression of tumor growth could be confirmed in an in vivo tumor model with nude mice. Mechanistically, DEFA5 directly bound BMI1, decreasing its binding at the CDKN2a locus, subsequently upregulating the expression of p16 and p19, two cyclin-dependent kinase inhibitors [107]. In contrast to this, DEFA5 and DEFA6 expression is significantly increased in colorectal cancer compared to healthy tissues. Moreover, DEFA5 may be associated with better prognosis of colorectal cancer, while DEFA6 may be linked to a worse prognosis. DEFA6 expression also seems to be increased significantly in adenoma compared to normal mucosa and slightly increased in carcinoma. More data is needed, but DEFA5 and DEFA6 seem to have a promising degree of specificity and sensitivity for predicting the prognosis of colorectal cancer [108].

For β-defensins, in general, hBD-1 seems to be downregulated in most tumor tissues, suggesting it has some form of anti-tumor activity. Datasets from patient cohorts show that hBD-1 transcription is decreased in colorectal cancer [109]. In human colon cancer cell lines and organoids of normal human colonic primary cells, inhibiting epidermal growth factor receptor (EGFR) increased the expression of hBD-1, whilst activating EGFR through the MAPK kinase 1/2 (MEKK1/2)-ERK1/2 pathway had the opposite effect [109]. Similarly, when liver cancer datasets were investigated for β-defensin expression, hBD-1 was significantly downregulated, with E-cadherin as the top positively correlated gene and hepatocyte growth factor-regulated tyrosine kinase substrate as the top negatively correlated gene, suggesting a protective role of this peptide in liver cancer development [110]. hBD-1 levels were also lower compared to healthy tissues in basal cell carcinoma, squamous cell carcinoma, renal cell carcinoma (promoting apoptosis), colon cancer and prostatic carcinoma [111,112,113,114,115,116,117]. This led to the speculation that hBD-1 is a tumor suppressor gene, and induction of its expression in prostate cancer cell lines indeed resulted in reduced cell growth [118]. In oral squamous cell carcinoma hBD-1 can suppress tumor migration and invasion and might act as a potential prognostic biomarker [119]. Downregulation of DEFB1 gene expression in oral squamous cell carcinoma correlates with worse prognosis, and according to a gene set enrichment analysis, its anti-tumor activity is associated with pathways impacting extracellular matrix remodeling, receptor tyrosine kinase (RTK)/ phosphoinositide 3-kinase (PI3K)/Akt/mechanistic target of rapamycin (mTOR) signaling, keratinization and cytokine signaling [120]. Interestingly, such anti-tumoral properties cannot be appointed to hBD-3, as its expression is upregulated in cervical carcinoma and hBD-3 even seems to promote cervical cancer cell growth by NF-κB signaling [121]. hBD-3 is also associated with fostering a tumor-promoting environment in human papillomavirus-associated head and neck cancer [122]. For hBD-1, its capacity to predict presence of prostate adenocarcinoma in morphologically normal prostate glands was investigated, in an attempt to lower false-negative biopsy cases. In 95.6% and 90.0% of prostate adenocarcinoma cases with, respectively Gleason Patterns 3 and 4, a loss of hBD-1 expression was observed. In a false-negative group, loss of hBD-1 expression in basal cells was a good biomarker for identifying high-risk patients with an initially negative biopsy [123]. The expression of hBD-1 to 4 is altered by cytostatic agents as well. In general, when added to different cancer cell lines, the tested cytostatic agents significantly suppressed the expression of hBDs, except for vincristine which caused a significant upregulation of hBD-1 and hBD-4 in a breast cancer cell line (MCF7) and doxorubicin which significantly upregulated the expression of hBD-3 and hBD-4 in an epidermoid carcinoma cell line (A431) [124]. 

Finally, when preparations of both α/β-defensins were administered to oral and oropharyngeal cancer patients receiving radiation therapy or chemotherapy, the defensins had an adjuvant effect, with a more pronounced effect on radiotherapy. The researchers claim that the use of defensins as an immune agent allows to achieve a more favorable tolerability profile of these standard anti-tumor therapies [125]. With regard to immunotherapy, physical exercise is known to be beneficial for your general health and immune system, and preliminary evidence supports the hypothesis that exercise would improve the immunological fitness of patients receiving immunotherapeutic regimens [126]. Taking this into account, it is interesting to note that in elite basketball athletes, HNP-1 and HBD-1 increased progressively with training intensification, while this increase in exercise does not change the levels of immune cells [127].

## 6. Conclusions

In conclusion, defensins are much more than just antimicrobial peptides. They can have an important role in the tumor microenvironment in two different ways. First, they have a direct cytotoxic effect on tumor cells, which is accompanied by a release of tumor antigens. Second, their role as immune mediator should not be ignored, as they chemoattract multiple types of immune cells, cause chemokine and cytokine release in the tumor microenvironment, and possibly mediate a variety of anti-tumoral immune processes through IFN-γ production, forming a bridge between the innate and the adaptive immunity.

## Figures and Tables

**Figure 1 ijms-24-05268-f001:**
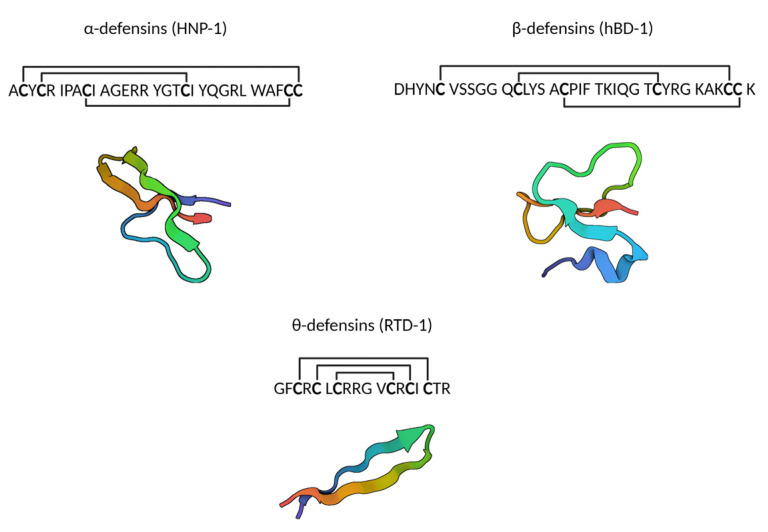
The three subfamilies of defensins, with an example sequence and accompanying structure. PDB codes of structures: 3GNY (HNP-1), 1E4S (hBD-1) and 2LYF (RTD-1) [23,24,25]. Created with BioRender.com.

**Figure 2 ijms-24-05268-f002:**
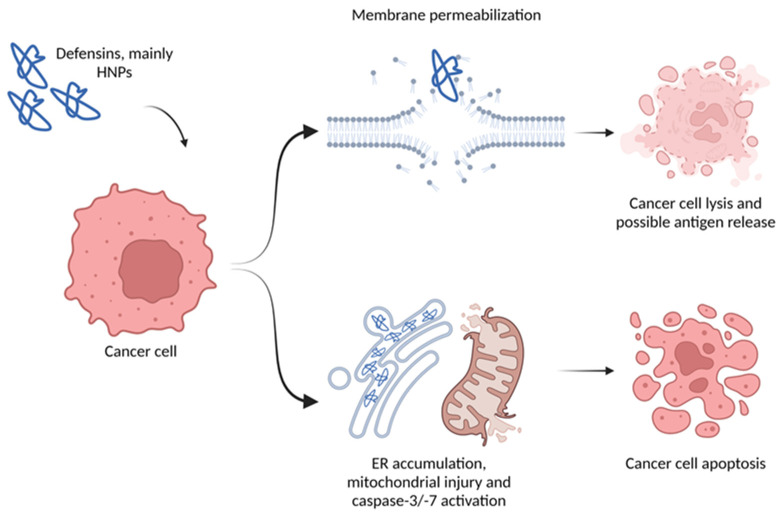
Direct effect of defensins on cancer cells. Therapeutic application of defensins can result in lysis or apoptosis. Created with BioRender.com.

**Figure 3 ijms-24-05268-f003:**
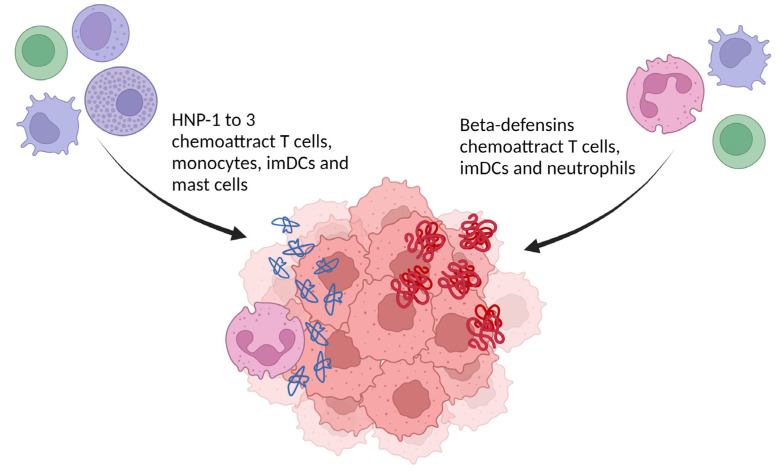
Chemoattractive effects of α- and β-defensins relevant in the tumor microenvironment. Those leukocytes can mediate an anti-tumoral immune response. (imDCs: immature dendritic cells). Created with BioRender.com.

**Figure 4 ijms-24-05268-f004:**
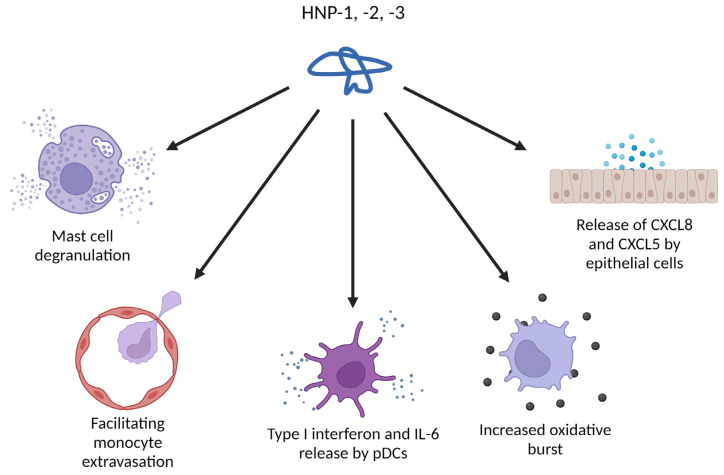
Immune-related activities of HNPs that can be relevant in the tumor microenvironment. Created with BioRender.com.

**Figure 5 ijms-24-05268-f005:**
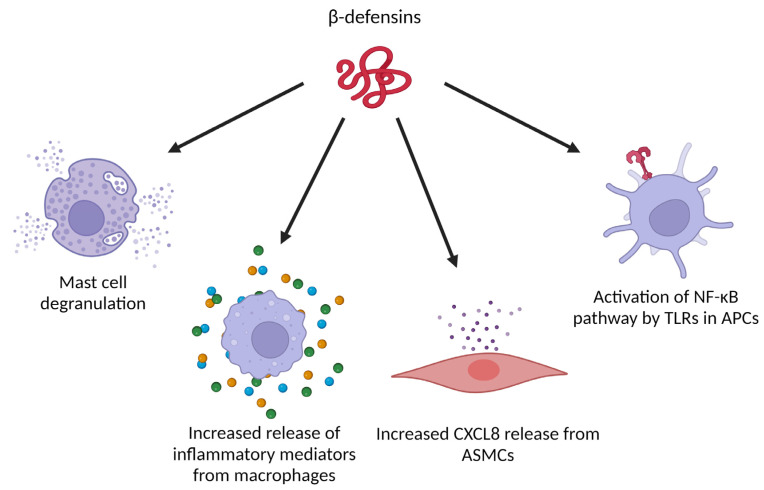
Immune-related activities of β-defensins that can be relevant in the tumor microenvironment. (ASMCs: airway smooth muscle cells, APCs: antigen presenting cells). Created with BioRender.com.

## Data Availability

No new data were created or analyzed in this study. Data sharing is not applicable to this article.

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
