# Peer review of "Role of Defensins in Tumor Biology"

_ijms, 2023, doi:10.3390/ijms24065268_

Round 1

Reviewer 1 Report

The manuscript is well-written and provides many details about the involvement of defensins in tumor biology. The Reviewer would only suggest adding some information about the interaction of defensins with complement cascade and its possible effect on tumor growth.

Author Response

A paragraph concerning this subject has been added to the manuscript. (lines 233-261)

Reviewer 2 Report

The authors of the proposed manuscript described the role of defensins in cancer biology.

The standard of the use of English is acceptable, and all the figures are present.

The manuscript has a typical shape with chapters entitled:

Page 1. 1. Introduction

Page 3. 2. Direct effect on tumor cells

Page 5. 2. Immune-related functions relevant in the tumor microenvironment

Page 8. 2. Proof of immune-stimulating effects in (animal) models

Page 9. 2. Expression of defensins in cancer

Page 10. 5. Conclusion

Could you explain why you used this order of numbers?

Line 318. According to The cancer genome atlas project, there is written “over 20,000 primary cancer and matched normal samples spanning 33 cancer types”. Please expand this paragraph in your proposed manuscript, taking into account all deposited cancer types.

Figure 1 lacks a source.

Figure 3 is unreadable. If the authors add a longer description in the main text (not only one sentence) or prepare a legend for the figure, It will be more understandable for the audience.

All of the figures have different fonts and are not sharp. Please standardize this.

In line 336 the authors wrote, “The expression level of α-defensin 5 336 (DEFA5) was drastically downregulated in human gastric cancer”. This sentence could be expanded by adding the newest results of the study in this field—for example, Zhao X 2023.

Author Response

Could you explain why you used this order of numbers?

Reply: This order of numbering was an error from our side. This has now been corrected.

Line 318. According to The cancer genome atlas project, there is written “over 20,000 primary cancer and matched normal samples spanning 33 cancer types”. Please expand this paragraph in your proposed manuscript, taking into account all deposited cancer types.

Reply: The paragraph has been expanded with taking the different cancer types into account. (lines 345-357)

Figure 1 lacks a source.

Reply: We included the references that describe the structures in the figure legend (legend of Figure 1).

Figure 3 is unreadable. If the authors add a longer description in the main text (not only one sentence) or prepare a legend for the figure, It will be more understandable for the audience.

All of the figures have different fonts and are not sharp. Please standardize this.

Reply: Figures have been standardized, all have the same font now and high resolution (300 DPI). Figure 3 is now readable and more understandable.

In line 336 the authors wrote, “The expression level of α-defensin 5 336 (DEFA5) was drastically downregulated in human gastric cancer”. This sentence could be expanded by adding the newest results of the study in this field—for example, Zhao X 2023.

Reply: This has been expanded with the new findings of Zhao X. (lines 376-383)

Round 2

Reviewer 2 Report

The authors addressed all of my concerns and question. 

The MS is ready to publish in its present form.